# FrontierBench: Are We Only Testing Agents Under the Streetlight?

## Abstract

As Large Language Models (LLMs) evolve into generalist agents capable of utilizing diverse tools, existing evaluation benchmarks are often confined to familiar search-type problems. It is crucial to search beyond the "streetlight" of known challenges and explore the "dark corners" where new capabilities are required. To address this gap, we first propose a new taxonomy (with 6 primary and 18 sub-problem types) of the LLM capabilities frontier, centered on the question: "Under what conditions do LLMs inherently fail, while tool-augmented agents can succeed?" Based on this, we introduce FrontierBench, a novel benchmark designed to evaluate generalist agents. We construct a multi-agent workflow that simulates a cognitive exploration process to generate testing problems. This workflow comprises three key stages: a cold-start step for directions, a targeted information gathering and environment preparing step, and an iterative question formulating step. Each stage incorporates an automated plan-action-replan sub-workflow, guided by our problem taxonomy to direct the exploration. Furthermore, we design a new metric, Knowledge Perplexity (K-PPL), which quantifies the novelty or "surprise" of new information in relation to what the LLM already knows and the current context. To generate more challenging problems, we run tool-restricted agents in parallel with our workflow. By comparing their relative progress (measured by K-PPL), a judge-LLM returns "descriptive rewards", steering the problem formulation towards more insightful information. Leading models like GPT-5 still fail ∼50% of execution tasks, even with advanced planning. Our FrontierBench offers a more realistic test of open-world potential.

## 1 Introduction

Recent years have witnessed a paradigm shift in Large Language Models (LLMs) (Dubey et al., 2024; DeepSeek-AI et al., 2025). Equipped with the ability to invoke external tools, they are transforming from passive text generators into intelligent agents capable of active planning and interaction with their environment (Li et al., 2023; Wentao Zhang, 2025; Qiu et al., 2025). This evolution enables them to search for dynamic information, execute code, operate software, and even engage with both physical and digital worlds, a capability that allows them to accomplish complex tasks previously beyond their reach. The Model Context Protocol (MCP) (Anthropic, 2024) serves as a standardized communication layer connecting LLMs with a vast array of tools, significantly lowering the barrier for developers to create and integrate tools for LLMs. This has fostered the growth of the tool ecosystem and expanded the LLMs' application boundaries.

However, faced with an ever-expanding suite of MCP tools and increasing task complexity, existing evaluation benchmarks have become inadequate. As illustrated in Figure Figure 1, classic benchmarks such as GAIA (Mialon et al., 2023), BrowseComp (Wei et al., 2025), and WebWalker-QA (Wu et al., 2025b) are largely confined to web search-oriented problems, with some works even artificially inflating difficulty by constructing lengthy and impractical queries. Meanwhile, emerging benchmarks like MCP-Bench (Wang et al., 2025), MCP-RADAR (Gao et al., 2025), and LiveMCP-101 (Yin et al., 2025) still have significant room for improvement in their coverage of problem types, tool categories, and the average length of decision-making chains. Consequently, we require a new benchmark to answer the question: What capabilities are exclusive to generalist agents and beyond the capability frontier of single LLMs?

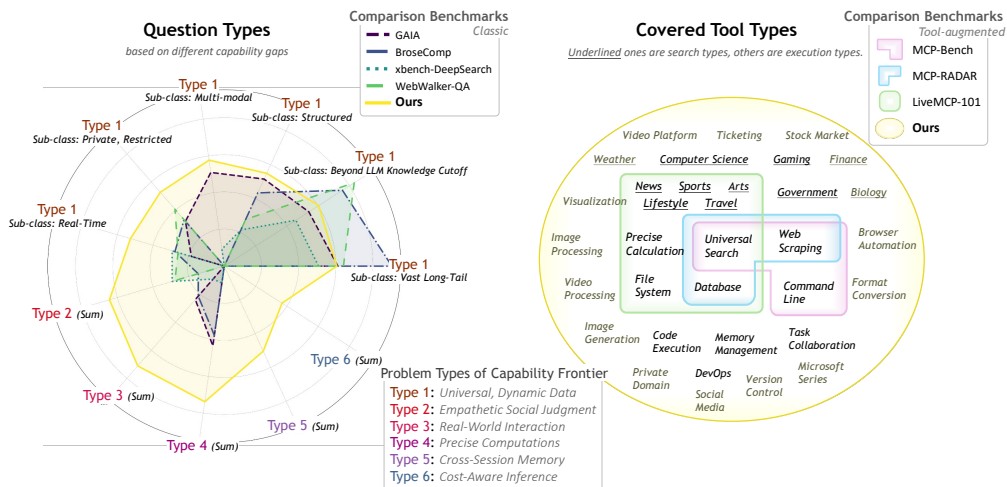

Figure 1: Comparison of FRONTIERBENCH with existing benchmarks on **question types** and **tool coverage**. The radar chart on the left indicates that FRONTIERBENCH (denoted as "Ours") achieves comprehensive and balanced coverage, whereas existing benchmarks like GAIA and WebWalker-QA primarily focus on information retrieval tasks. The right part shows that FRONTIERBENCH also covers a broader range of tool types than recent MCP-tool-augmented benchmarks.

To address this, we first propose a novel agent capability taxonomy. We systematically categorize the value of tools as bridging six distinct *capability frontier* for LLMs, including 1) universal, dynamic data, 2) empathetic social judgment, 3) real-world interaction, 4) precise computation, 5) cross-session memory, 6) cost-aware inference. This taxonomy provides a structured foundation for understanding how tools expand an LLM's inherent limitations.

Building on this framework, we introduce **FrontierBench** and present an automated workflow for constructing testing tasks. This workflow simulates an agent's cognitive exploration process to generate high-quality, structurally complex tasks. It begins with **1**) a **cold-start phase** to find diverse exploration directions, ensuring initial seeds are not those that LLMs naturally excel at. This is followed by **2**) a targeted **information gathering** and **environment preparation phase**, where we acquire related data and setups for problem construction, then structure it into a Knowledge Directed Acyclic Graph (DAG). Finally, in **3**) an **iterative question formulation stage**, the agent generates and validates solvable sub-problems based on the information consolidated in the Knowledge DAG. Each stage is driven by an automated plan-action-replan sub-workflow. A **Planner Agent** formulates plans, an **Action Agent** executes sub-tasks by invoking toolsets, and a **Context Manager** prunes redundant information, summarizes key findings, and updates the Knowledge DAG. This allows the planner to leverage the continuously enriched knowledge graph to refine its subsequent plans.

To measure the effectiveness of this exploration process, we introduce **Knowledge Perplexity (K-PPL)**, a metric inspired by linguistic perplexity. By computing the conditional probabilities between key information nodes in the Knowledge DAG, K-PPL aims to interpret the "cognitive state" in real-time. When the agent encounters novel information that its existing knowledge cannot explain, its cognitive uncertainty increases, causing K-PPL to rise. Conversely, when new information successfully connects and explains existing knowledge points, leading to "aha" moments, and K-PPL falls. The trajectory of K-PPL reveals how our process avoids the streetlight effect, the tendency to familiar problems, by pushing agents into the challenging "dark corners" of the problem space.

To discover problems that are even more challenging, we enhance our workflow with an adversarial form guided by K-PPL. In this framework, we run a tool-restricted (i.e., weaker) agent in parallel with our primary exploration one. A separate judge-LLM monitors both agents, comparing their relative progress based on success rates and K-PPL dynamics. When the primary agent succeeds or makes significant cognitive progress (indicated by K-PPL changes) on a task where the weaker agent fails, the judge provides a descriptive reward. This reward guides the creation of challenging problems that can distinguish between agents of different capability levels. In summary, our contributions are:

Figure 2: **The taxonomy of our problem type from the agent capability frontier**, by the types that necessitate tool-augmented agents and are beyond a single LLM.

- A novel agent capability taxonomy, structured around six *capability frontiers*, which provides a foundation for building comprehensive and in-depth benchmarks.
- An automated data construction workflow that simulates cognitive exploration through multi-agent collaboration, generating complex evaluation tasks complete with their full exploration paths.
- The introduction of Knowledge Perplexity (K-PPL) to interpret knowledge exploration behavior and an adversarial workflow driven by a K-PPL-based descriptive reward, which runs in parallel with weaker agents, to generate evaluation data at the capability frontier.

## 2 PRELIMINARY

**Motivation:** Our taxonomy is rooted in the fundamental limitations of the LLM paradigm. By their very design, LLMs are static models trained on vast but offline datasets. This architecture inherently prevents them from accomplishing many real-world problems.

An agent-based approach, which equips LLMs with external tools with context management, is essential for surmounting these constraints. To systematically frame this necessity, we propose a taxonomy that categorizes the value of generalist agents. As illustrated in Figure 2, this taxonomy identifies the six core capability gaps that agents should bridge, defining the frontier beyond which an LLM cannot operate.

- **Universal, Dynamic Data:** For the real-time, dynamic information, *e.g.*, weather, stock prices, that changes at high frequency and whose value decays rapidly; that is updated after the LLM's training cutoff date; long-tail details, private, or domain-specific data.
- **Empathetic Social Judgement:** LLMs are unbiased without subjective experiences. When faced with situations with personal feelings, dynamic cultural contexts, or complex ethical dilemmas, LLMs cannot independently make the value-laden judgments that align with human values.
- **Real-World Interaction:** LLMs cannot directly interact with the physical world or complex software environments. This prevents them from executing procedural skills, perceiving physical states, or performing closed-loop control that requires real-time feedback.
- **Precise Computation:** The probabilistic, auto-regressive nature of LLM generation makes them unreliable for tasks requiring high-precision mathematical calculations, rigorous formal logical reasoning, or deterministic algorithmic execution.
- **Cross-Session Memory:** Constrained by a finite context window, LLMs lack coherent memory across long periods and multiple interactions, making it difficult to track the progress of complex tasks or to dynamically build and evolve user profiles.

Table 1: **Related works:** Comparison of Mainstream Generalist Agent Benchmarks

| | Diversity | | | Difficulty | | |
|---|---|---|---|---|---|---|
| | Exec. | Mem. | Search Sol. | Levels | Construct. | Evaluat. |
| *Generalist Agent Bench* | | | | | | |
| **GAIA** | ✗ | ✗ | 90% | 3, by info. nodes | Human-created | Veri. result |
| **BrowseComp** | ✗ | ✗ | 95% | Not mentioned | Human-created | Veri. result |
| **xbench-DS** | ✗ | ✗ | 95% | 4, by est. human expert time | Industry collected | Veri. result |
| **HLE** | A few | ✗ | 65% | Not mentioned | Human-collected, LLM-filtered, Expert-reviewed | Veri. result, Error, Cost |
| **WebWalker-QA** | ✗ | ✗ | 98% | 3, by info. page depth | Web crawled, Synthesized | Veri. result, Cost |
| *Latest Bench w/ MCP Tools* | | | | | | |
| **MCPEval** | A few | ✗ | 70% | Not mentioned | LLM-gen., iteratively refined | Tool-call acc., Completion rate by LLM |
| **MCPBench** | Very few | ✗ | 85% | Not mentioned | Sampled, Synthesized from existing data | Veri. result, Cost |
| **MCP-RADAR** | Very few | ✗ | 75% | 3, by tool-calls | Rewritten from existing data, Human-revised | Veri. result, Tool-call acc., First error pos., Cost |
| **LIVEMCP-101** | ✓ | ✗ | 60% | 3, by difficulty | LLM-gen., Human Revised | Parallel exec. w/ tool-call-GT |
| **FrontierBench(Ours)** | ✓ | ✓ | **50%** | **3, by info. nodes** | **Automated workflow, Human-revised** | **Veri. result (if have), Parallel exec. w/ info. nodes** |

- **Cost-Aware Inference:** When generating decisions, LLMs lack awareness of real-world resource costs, such as API call fees, computation time, or budgets, potentially leading to solutions that are effective but prohibitively expensive.

This taxonomy provides first-principles guidance for constructing a comprehensive benchmark.

## 3 RELATED WORKS

**Agent Workflow** Related works build upon the "LLM + Tools" foundation to further optimize Planning, Memory, and Collaboration patterns. For example, Auto-GPT (Significant-Gravitas, 2023) and Tree of Thoughts (Zhou et al., 2024) focus on planning and decomposition, enhancing execution reliability by breaking down complex tasks into manageable sub-steps. Research such as Reflexion (Shinn et al., 2023) centers on memory and self-reflection, enabling agents to learn from past experiences and dynamically adjust their behavior. Furthermore, frameworks like AutoGen (Wu et al., 2023) and MetaGPT (Hong et al., 2024) explore multi-agent collaboration, simulating human teams by assigning specialized roles to solve complex problems that are intractable for a single agent.

These complex agent frameworks can be unified under the paradigm of "LLM + Tools + Sophisticated Context Management." For instance, role-playing in multi-agent systems can be viewed as a context partitioning strategy. By assigning distinct roles to different agents, the global context is decomposed into multiple, more focused sub-contexts, reducing the cognitive load on any single LLM. However, there is a distinction between the design goal of these frameworks (solving problems within a given space) and the goal of constructing high-quality evaluation questions.

**Workflow for Constructing Questions** Existing methods for constructing evaluation questions are somewhat limited. Some approaches (*e.g.*, E2HQA (Wu et al., 2025a), WebShaper (Tao et al., 2025)) increase difficulty by adding constraints to existing problems, but this often results in queries that are verbose and impractical. Other methods (*e.g.*, CRAWLQA (Wu et al., 2025a)) use a iterative-walk approach to gather information for question generation, but this is prone to the "street-light effect," where it only focuses on information that is easy to find and evaluate.

## 4 BUILDING FRONTIERBENCH BEYOND THE STREETLIGHT

**Motivation:** Manual methods to construct a comprehensive benchmark are costly and often yield static question-answer pairs that fail to capture the dynamic, trial-and-error nature of real-world problem-solving. Conversely, simple automated generation methods are susceptible to the "Streetlight

Effect", the tendency to generate problems within an LLM's existing comfort zone, failing to produce tasks with sufficient complexity and novelty to probe its capability boundaries.

To overcome these limitations, we designed an automated cognitive exploration workflow. This process mimics human problem-solving by iteratively **exploring** and **validating**. Starting from a conceptual "seed," it leverages tools to uncover *unknown* information and progressively builds a multi-step reasoning chain, culminating in a complex, deep-seated problem.

### 4.1 AUTOMATED COGNITIVE EXPLORATION

Our workflow systematically executes the three stages outlined in Figure 3: **1**) a cold-start phase for direction finding, **2**) an information gathering and environment preparation phase, and **3**) an iterative question formulation phase.

#### 4.1.1 MAIN WORKFLOW

The main workflow unfolds across three stages. It begins with the **Cold-Start Phase** to ensure the breadth and novelty of the generated problems. Its primary goal is to produce a diverse set of orthogonal "seed" directions, steering the exploration process away from topics that LLMs can easily solve. Next, in the **Information Gathering and Environment Preparation Phase**, the system lays the groundwork for a solvable yet complex task. For a given seed, the agent autonomously collects relevant data, prepares necessary environmental states, and organizes these assets into an initial **Knowledge Directed Acyclic Graph (Knowledge DAG)**. Finally, the **Iterative Question Formulation Phase** synthesizes the sub-problems on the Knowledge DAG into a final, complex question complete with a verifiable solution path.

Each stage is driven by a unified, closed-loop sub-workflow where a **Planner Agent**, an **Action Agent**, and a **Context Manager** work in synergy. This system continuously interacts with the environment via a toolset to dynamically build and maintain a Knowledge DAG, which serves as its structured memory and reasoning canvas.

#### 4.1.2 CORE COLLABORATIVE SUB-WORKFLOW

For any given objective within a stage, the system initiates a dynamic, Knowledge DAG-driven exploration cycle:

1. **Plan Generation**: The **Planner Agent** formulates an initial strategy based on the available tools. It decomposes the objective into a series of sub-tasks, which form the initial nodes $\mathcal{N}$ in a Knowledge DAG, $\mathcal{G} = (\mathcal{N}, \mathcal{E})$. Each node represents a sub-problem, its current state, and its dependencies as $\mathcal{E}$.
2. **Task Execution**: The **Action Agent** executes the sub-tasks dispatched by the Planner. Following the ReAct framework, it interacts with the environment through a "thought-action-observation" cycle, invoking the necessary tools. This execution creates a micro-exploration trajectory, $\tau_j = (a_1, o_1; a_2, o_2; \dots)$, which captures the sequence of tool calls and their outputs leading to the completion of sub-task $n_j$.
3. **Knowledge Integration**: The **Context Manager** acts as the system's memory hub. It prunes redundant data from the trajectory $\tau_j$, extracts key findings, and integrates this structured information back into the Knowledge DAG. This keeps the graph concise and focused on solution-relevant knowledge.
4. **Dynamic Re-planning**: The **Planner Agent** reviews the updated Knowledge DAG. Based on newly discovered nodes (unexpected findings), validated paths (successful sub-tasks), or dead ends (failed attempts), it dynamically refines its overarching plan. It can access the raw trajectory data of any node to:

   - **Deepen**: Formulate more advanced insights along a promising path.
   - **Broaden**: Initiate a new exploratory branch from a key discovery.
   - **Correct**: Prune an ineffective path and backtrack to a more promising state.

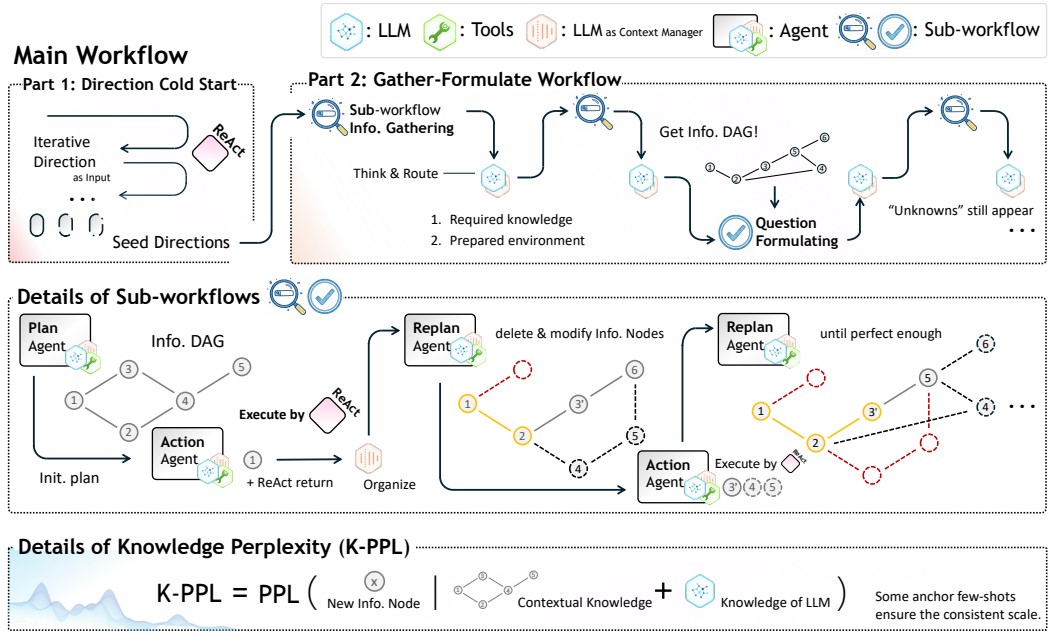

Figure 3: **An overview of the automated cognitive exploration workflow** for constructing FRON-TIERBENCH problems. The "Main Workflow" at the top is divided into a "cold-start" phase for exploring directions and a "gather-formulate" phase for evaluation questions. The middle section illustrates the iterative "plan-action-replan" sub-workflow for each stage.

This "plan-execute-integrate-replan" cycle iterates until the stage's objective is met. The final output is a highly structured data instance containing a complex evaluation problem, its golden solution path, and a rich context graph recording all exploration attempts.

**More Details:** To maximize the effectiveness, we incorporate two key preparatory steps:

- **Preliminary Tool Exploration:** Before task generation, an agent performs preliminary trials on tool subsets to summarize each tool's core function, ideal use cases, and limitations. This "tool manual" serves as a crucial reference, improving the Planner's decision-making and the Action Agent's execution efficiency.
- **Orthogonal Seed Generation:** To ensure the benchmark's breadth and novelty, the cold-start phase is designed to produce a diverse set of orthogonal seed directions. By combining tool capabilities with their corresponding capability frontiers, we iteratively generate seeds. During this process, already generated topics are injected back into the prompt, explicitly instructing the model to propose new directions, thereby maximizing conceptual coverage and avoiding overlap.

## 4.2 GUIDANCE BASED ON KNOWLEDGE PERPLEXITY (K-PPL)

To objectively quantify the effectiveness of the cognitive exploration process and to verify that it does not fall into an inefficient random walk characteristic of the "Streetlight Effect," we introduce **Knowledge Perplexity (K-PPL)**. K-PPL is designed to monitor the "cognitive state" of the data-constructing agent in real-time. Its core hypotheses are:

- When the agent encounters novel information that its existing knowledge system cannot easily explain, its cognitive uncertainty increases, causing K-PPL to **rise**. This may signal the discovery of a potentially complex problem or a creative challenge.
- When the agent acquires information that effectively connects previously isolated knowledge points, its cognitive uncertainty decreases, causing K-PPL to **fall**. This may correspond to an "Aha!" moment of forming a coherent explanation.

**K-PPL Definition and Calculation.** At any timestep $s$, the Knowledge Perplexity of the Knowledge DAG $G_t = (\mathcal{N}_t, \mathcal{E}_t)$ is defined as:

$$\text{K-PPL}(G_t) = \Pr(G_t)^{-\frac{1}{N_s}}, \quad \text{where } \Pr(G_t) = \prod_{j=1}^{N_s} \Pr(n_j \mid \text{Prev}(n_j))$$

Here, $N_s$ is the total number of nodes in the DAG. The conditional probability of a single node $n_j$ is decomposed into a weighted sum of **External Relevance** and **Internal Prior**:

$$\Pr(n_j \mid \text{Prev}(n_j)) = \omega \cdot \text{Rel}(n_j, \text{Prev}(n_j)) + (1 - \omega) \cdot \text{Prior}(n_j)$$

For the root node, $\omega = 0$. We use an LLM identical to the one constructing the data to estimate these values, providing few-shot examples to anchor the scale:

- **Relevance Score (Rel)**: Measures the explanatory or predictive strength of the parent nodes' information set on $n_j$.
- **Prior Score (Prior($c_i$))**: Measures the commonness or unexpectedness of the information $n_j$ itself, i.e., the probability that the LLM knows the concept without any context.

The dynamic evolution of K-PPL is driven by three atomic operations on the Knowledge DAG: Add Node, Add Edge, and Update Node Information, all of which trigger a recalculation of conditional probabilities. We found that simple exploration strategies tend to exhibit *lazy convergence*, characterized by a K-PPL curve that drops sharply initially and then becomes excessively smooth without significant spikes. While seemingly efficient, this behavior stifles the potential for discovering deeper problems. In contrast, a general downward trend interspersed with sharp spikes is indicative of an exploration process more likely to yield high-quality problems.

We emphasize that not all increases in K-PPL are beneficial; erroneous tool outputs, model hallucinations, or irrelevant information can also cause it to rise. Therefore, the instantaneous fluctuation of K-PPL is not the sole criterion for judgment. It should be evaluated in conjunction with historical trends, context, and the agent's overall cognitive state to understand the reasons behind the changes.

## 4.3 Adversarial Generation Workflow with Comparative K-PPL

To discover tasks that probe the harder capability frontier, we introduce an adversarial workflow. This process refines the difficulty of generated tasks by comparing the performance of our primary exploration agent against a set of weaker agents. These weaker agents operate in parallel with the primary one, but their toolsets are strategically restricted according to our agent capability taxonomy.

The core of this process is governed by a judge-LLM. When the primary agent's exploration sub-workflow establishes a new information node, this node is framed as a validation subtask for all agents to attempt. The judge-LLM then assesses their relative performance by analyzing their success rates and K-PPL dynamics. A task is deemed challenging if the weaker agents either fail to solve it or exhibit "struggling" K-PPL trajectories, *e.g.*, unresolved perplexity spikes or significantly higher path perplexity, while the primary agent succeeds.

Based on this comparison, the judge-LLM provides a **descriptive reward** to the Planner Agent guiding the primary exploration:

- **High-Difficulty Feedback:** Positive feedback is given when a task effectively distinguishes the primary agent from its weaker counterparts. This encourages the planner to deepen its exploration along the current, challenging path.
- **Low-Difficulty Reward:** Constructive feedback is provided if the task is easily solved by the weaker agents. This indicates the exploration is too superficial, prompting the planner to pivot towards more complex problem domains.

This adversarial loop creates an intrinsic, difficulty-driven curriculum. By continuously challenging the primary agent to generate tasks that stump progressively more capable versions of the weaker agents, our workflow ensures the resulting FRONTIERBENCH problems truly reside at the edge of current agent capabilities.

Table 2: Performance of Different Models and Workflows on FRONTIERBENCH

| Workflow | Model | Level 1 | | | Level 2 | | | Level 3 | | |
|---|---|---|---|---|---|---|---|---|---|---|
| | | Acc. | Paral. Acc. | Cost | Acc. | Paral. Acc. | Cost | Acc. | Paral. Acc. | Cost |
| ReAct | GPT-4o$_{mini}$ | 72.2 | 65.8 | 1830 | 66.1 | 61.9 | 2504 | 33.3 | 25.5 | 3198 |
| | GPT-4o | 76.1 | 70.3 | 2473 | 68.9 | 66.5 | 3850 | 33.3 | 31.9 | 4927 |
| | GPT-5 | 94.8 | 97.3 | 4824 | 87.2 | 89.3 | 9601 | 55.6 | 66.0 | 14505 |
| SmolAgent | GPT-4o$_{mini}$ | 69.1 | 66.7 | 1417 | 70.0 | 63.5 | 2485 | 27.8 | 36.2 | 2797 |
| | GPT-4o | 82.2 | 75.2 | 2373 | 68.3 | 65.5 | 4163 | 33.3 | 31.9 | 4651 |
| | GPT-5 | 97.8 | 95.5 | 4679 | 86.7 | 89.8 | 8240 | 55.6 | 61.7 | 17380 |
| OWL | GPT-4o$_{mini}$ | 73.0 | 68.9 | 2440 | 72.2 | 65.5 | 3825 | 36.1 | 42.6 | 4474 |
| | GPT-4o | 83.9 | 73.4 | 2881 | 68.3 | 66.5 | 4805 | 30.6 | 27.7 | 5573 |
| | GPT-5 | 97.0 | 99.1 | 5262 | 90.6 | 87.8 | 13298 | 50.0 | 57.4 | 18265 |

## 5 EXPERIMENTS

We evaluate FRONTIERBENCH to assess the performance of state-of-the-art models, quantify the effectiveness of our Knowledge Perplexity (K-PPL) metric in capturing cognitive exploration, and validate our K-PPL-guided adversarial generation workflow for creating high-difficulty tasks.

**Experimental Setup** Evaluations are conducted on FRONTIERBENCH, with questions categorized into three difficulty levels based on the number of nodes in their knowledge DAG: Level 1 (Easy), Level 2 (Medium), and Level 3 (Hard). We test three base models: **GPT-4o-mini**, **GPT-4o**, and **GPT-5**. These models are integrated into three representative agent workflow frameworks: **ReAct**, a baseline "thought-action-observation" paradigm; **SmolAgent**, which generates Python code snippets for actions to improve efficiency; and **OWL** (Optimized Workforce Learning), a hierarchical multi-agent framework that decouples planning from execution for enhanced cross-domain transferability.

We employ two primary metrics. **Accuracy (Acc.)** is determined by an LLM-as-Judge comparing agent outputs to a ground truth or predefined criteria. For tasks dependent on dynamic states, we introduce **Parallel Accuracy (Paral. Acc.)**, where a test agent's execution trajectory and key state discoveries are compared against those of an "Oracle LLM" possessing the golden solution path. This offers a more robust measure of decision-making in dynamic environments.

### 5.1 MAIN RESULTS AND ANALYSIS

#### 5.1.1 OVERALL PERFORMANCE OF AGENTS ON FRONTIERBENCH

As shown in Table 2, task difficulty is the decisive factor in agent performance. Accuracy drops precipitously across all configurations as difficulty increases from Level 1 to Level 3; for instance, GPT-5 with ReAct falls from 94.8% to 55.6%. This highlights FRONTIERBENCH's success in creating tasks that challenge even top-tier models, exposing their limitations in long-term planning and precise environment interaction, especially on execution-type questions (Figure 5, left). While more sophisticated workflows like SmolAgent and OWL offer marginal improvements, the primary bottleneck remains the LLM's core reasoning capabilities, not the external framework. Furthermore, the results reveal a trade-off between cost and performance, as higher difficulty levels demand exponentially more computational resources, where smaller models show significant limitations.

#### 5.1.2 ANALYSIS OF K-PPL'S ROLE IN THE COGNITIVE EXPLORATION PROCESS

Figure 4 visualizes how K-PPL captures an agent's cognitive process. A standard ReAct workflow exhibits the "Streetlight Effect," where its K-PPL curve rapidly smooths out, indicating it remains within its knowledge comfort zone. In contrast, our workflow demonstrates a "deep exploration" mode, characterized by a downward trend interspersed with sharp K-PPL spikes. These spikes signify encounters with novel information that increases cognitive uncertainty, which is then resolved as the agent integrates the knowledge, causing K-PPL to fall in an "Aha!" moment. This pattern confirms that our process actively seeks and resolves cognitive conflicts to discover complex problems.

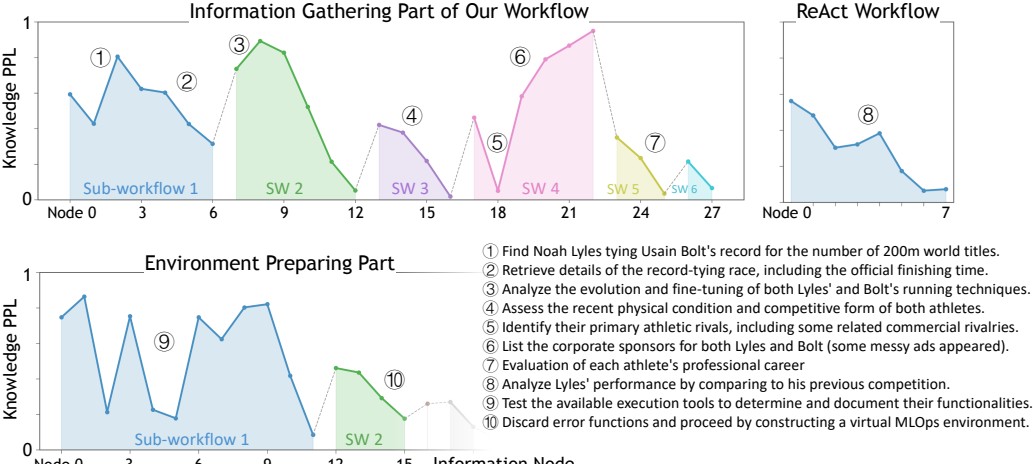

Figure 4: **A comparative analysis of K-PPL trajectories.** The left chart displays the K-PPL dynamics of our proposed workflow during the "information gathering" and "environment preparing" stages. The right chart shows the K-PPL curve for a standard ReAct workflow.

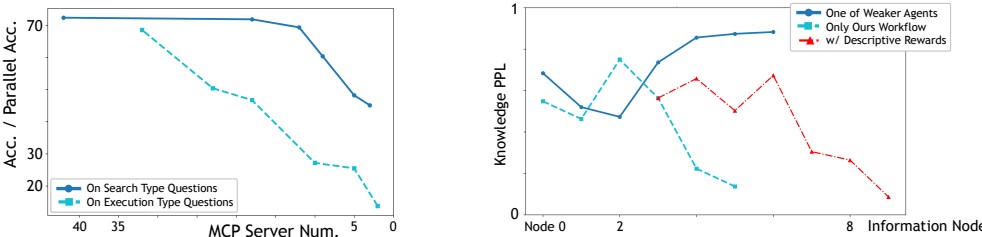

Figure 5: **Validation of the effectiveness** of the K-PPL-guided adversarial generation workflow. The left illustrates the tool dependency of different problem types. The right visually demonstrates how the descriptive rewards guides the subsequent exploration (red line) to become more adventurous.

### 5.1.3 EFFECTIVENESS OF K-PPL-BASED ADVERSARIAL GENERATION

The effectiveness of our K-PPL-guided adversarial workflow is demonstrated in Figure 5. The left panel validates the design of execution-type tasks by showing their stronger dependency on tool availability compared to search-type questions. The right panel provides direct evidence of adversarial guidance. When a tool-restricted weaker agent struggles at an information node, the Judge-LLM observes that our main workflow resolves it successfully and issues a descriptive reward. After receiving this reward, the main workflow's exploration strategy becomes more "adventurous," exhibiting a more dynamic K-PPL curve with alternating peaks and valleys. This shows the reward successfully incentivizes the planner to explore more challenging domains, ensuring the generated tasks are both difficult and discriminative.

## 6 CONCLUSION & LIMITATION

We introduce FRONTIERBENCH, a novel agent benchmark designed to overcome the "streetlight effect." Guided by a new capability taxonomy and a Knowledge Perplexity metric, our automated, multi-agent workflow generates complex tasks that probe capability frontiers. Experiments demonstrate that FRONTIERBENCH effectively reveals the limitations of even state-of-the-art LLMs like GPT-5, particularly in complex planning and interaction, with their accuracy dropping to approximately 55% on the most difficult tasks. A potential limitation of our work is the significant resource consumption associated with the adversarial generation process.

## ETHICS STATEMENT

We have strictly adhered to the ICLR Code of Ethics. Our research does not involve any human subjects, sensitive data, or personally identifiable information. The work presented in this paper does not raise concerns regarding discrimination, bias, or potential for malicious use. We have conducted our research with integrity and are committed to the responsible advancement of machine learning.

## REPRODUCIBILITY STATEMENT

We are committed to ensuring the reproducibility of our research. All implementation details, model architectures, and hyperparameters are thoroughly described in the main body of the paper and the appendix. To further facilitate reproducibility, we will make our complete source code and experimental scripts publicly available in a GitHub repo. upon the acceptance.

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
