# OpenReview forum: "FrontierBench: Are We Only Testing Agents Under the Streetlight?"
_ICLR.cc/2026/Conference — Submitted to ICLR 2026_

### Official Review · Reviewer_CPMr · 2025-10-15

**Soundness:** 1
**Presentation:** 4
**Contribution:** 2
**Rating:** 2
**Confidence:** 4

**Summary:**

This paper introduces FrontierBench, a new benchmark meant to test large language models (LLMs) and tool-using agents on tasks that go beyond the usual search or question-answering problems. The authors argue that current benchmarks only test what LLMs already do well (“under the streetlight”) and miss more realistic or open-ended challenges.

They first define a taxonomy which they then use to build a multi-agent system that generates new benchmark problems.

A new metric (K-PPL) measure how surprising/novel new information is, based on what the knowledge already knows and the context. The system uses this to guide exploration and create harder tasks, which they succeed in: GPT-5 fails on a large percentage of the newly generated tasks. The authors argue this makes it a more realistic test of true open-world capabilities.

**Strengths:**

The creative framing is excellent, the "streetlight" metaphor is a great intuition pump to quickly understand the rationale/goal of the paper, which I agree is highly relevant.

The K-PPL metric is conceptually very interesting, I would really like to read a paper focussing slowly on that, validating it on already published problems, rather than on a custom framework (FrontierBench).

The multi-agent generation workflow is well-detailed and technically sophisticated.

**Weaknesses:**

The benchmark design involves many in my opinion arbitrary choices, especially the taxonomy categories, but also the DAG-based structure, parameterization of K-PPL, definition of "difficulty" that appear more heuristic than principled.

The criticism of other benchmarks "artificially inflating difficulty" is not convincing without further argument. The same critique could apply to FrontierBench's own adversarial generation.

The paper attempts too many contributions at once, taxonomy, framework, adversarial workflow, and a novel metric, without any one being deeply validated.

The evaluation of K-PPL remains limited. It is unclear how it behaves on existing benchmarks (which would be the more important comparisons), across models, or whether it correlates with intuitive notions of task novelty or difficulty.

Conceptually strong, but experimental interpretation is more anecdotal than systematic.

I don't think it is hard to come up with problems for which GPT-5 fails, which is used several times as an indication that coming up with difficult tasks works.

**Questions:**

Why was K-PPL defined using a Knowledge DAG rather than simpler formulations? Did you try other approaches?

How does K-PPL correlate with task difficulty on established benchmarks? Does it work as expected when comparing models with different training corpora?

How sensitive is FrontierBench to design choices, especially the taxonomy, or the structure of the judge-LLM’s descriptive rewards?

To what extent are the generated tasks truly novel vs. synthetic recombinations of existing domains? How do they really differ from the "artificially inflated" difficulty of standard benchmarks, other than encompassing broader tools?

Which parts do you as authors find to be the more valuable contribution? K-PPL, or the task generation pipeline? Which benefits of K-PPL beyond the FrontierBench frameworks do you see?

---

### Official Review · Reviewer_qZ31 · 2025-10-30

**Soundness:** 3
**Presentation:** 2
**Contribution:** 2
**Rating:** 2
**Confidence:** 4

**Summary:**

The paper proposes FRONTIERBENCH, a new benchmark for evaluating the frontier capabilities of LLM-based agents beyond conventional search-style tasks. It introduces a six-dimensional taxonomy of LLM limitations and a three-stage automated workflow that simulates cognitive exploration through planning, acting, and re-planning with a shared knowledge graph. A novel metric, Knowledge Perplexity (K-PPL), quantifies the novelty of discovered information and guides an adversarial generation process where stronger and weaker agents are compared to create increasingly challenging tasks. Experiments on GPT-4o-mini, GPT-4o, and GPT-5 show that even the strongest models fail on about half of the hardest problems, suggesting that FRONTIERBENCH effectively probes the limits of current agent capabilities.

**Strengths:**

1. The paper tackles a timely and important question with well-articulated and meaningful motivation.
3. The automated cognitive exploration workflow, with its plan–action–replan cycles and Knowledge DAG structure, is solid.
4. The proposed K-PPL metric is interesting.
5. The paper is overall well-structured, with strong visualizations that clarify both the workflow and taxonomy.

**Weaknesses:**

- The paper omits key details regarding benchmark construction and dataset composition. It is unclear how many tasks exist per capability type or difficulty level, and what exact thresholds define “Level 1–3” in terms of Knowledge DAG nodes.
- Quantitative statistics such as the total number of generated tasks or category balance are missing, making it hard to evaluate the benchmark’s comprehensiveness.
- The workflow consists of three stages, each described as an independent exploration process, but their differences and concrete designs are insufficiently explained.
- The mechanism ensuring that initial seeds escape the “streetlight” region is not guaranteed or validated. Since the seeds are still generated by an LLM, the process may inherently bias exploration toward familiar domains, limiting the discovery of truly novel challenges.
- The pruning and updating strategy of the Knowledge DAG is under-specified: it is unclear how redundant or noisy nodes are defined, detected, or pruned during iterative updates.
- The proposed Knowledge Perplexity (K-PPL) is conceptually appealing but methodologically uncertain. It is computed using another LLM, which raises concerns of circularity and bias.
- All experiments are conducted using GPT-family models (GPT-4o-mini, GPT-4o, GPT-5). This narrow evaluation scope limits conclusions about cross-model robustness.
- The paper would benefit from clear, end-to-end illustrative examples contrasting a “streetlight” task from existing benchmarks with a “frontier” task generated by FRONTIERBENCH to concretely demonstrate its claimed novelty. Right now, it remains unclear whether the benchmark truly explores the “dark side” of the capability frontier, rather than simply generating harder or more failure-prone tasks.

**Questions:**

See above.

---

### Official Review · Reviewer_FcJS · 2025-10-31

**Soundness:** 1
**Presentation:** 1
**Contribution:** 1
**Rating:** 2
**Confidence:** 4

**Summary:**

This paper seeks to create a benchmark for LLM agents equipped with tools that goes beyond the "streetlight" of existing benchmarks. To do so, it considers a taxonomy of frontier abilities for agents, including universal dynamic data, social judgements, real-world interaction, precise computations, cross-session memory, and cost-aware inference. Rather than manually curating data, the paper has agents "iteratively explore and validate" starting from a "conceptual seed." The whole approach is premised on the reliability of LLM-as-a-judge. Finally, a "knowledge perplexity" metric is introduced to "quantify the effectiveness of the cognitive exploration process."

**Strengths:**

- I agree with the high-level premise of the paper: that dynamic benchmarks that test a diverse set of skills are important.
- The proposed taxonomy of abilities is reasonable.
- The K-PPL trajectories seem to have some interesting properties (but see weaknesses).

**Weaknesses:**

- There are no clear research questions articulated in the paper.
- It seems like there are many design decisions in the "workflow" used, which amount to model selection. However, I'm unclear on whether a separate validation dataset was used to set these hyperparameters.
- The results seem dependent on a wide range of engineering details (e.g., prompts, sampling configuration, etc.), many of which are not detailed in the paper.
- Insufficient details are provided about the actual problems being solved. There need to be examples early on providing concrete grounding for what the paper is about.
- It is quite unclear where there ground truth labels come from. Are these gold references? Silver labels? How were they obtained?
- The definition of K-PPL is imprecise. Provide an example of how this is calculated.
- Overall, the technical contributions are lacking; I haven't learned anything from reading this paper.
- The discussion of limitations is poor.
- The reproducibility statement is inaccurate; many details are missing.

**Questions:**

See "Weaknesses"

---

### Official Review · Reviewer_nHLv · 2025-11-01

**Soundness:** 1
**Presentation:** 1
**Contribution:** 2
**Rating:** 2
**Confidence:** 4

**Summary:**

The paper proposes FrontierBench, a benchmark that targets “capability frontiers” where agents face dynamic data, social judgment, real-world interaction, precise computation, cross-session memory, and cost-aware inference. It builds tasks through a multi-agent plan–act–replan workflow that constructs a Knowledge DAG and then measures exploration with a Knowledge Perplexity metric called K-PPL. The authors claim broad coverage across question and tool types and show example K-PPL trajectories to argue the workflow finds harder problems. However, the paper presents FrontierBench largely at a conceptual level and key operational details are missing, which limits trust and reproducibility.

**Strengths:**

1. The idea is relevant. Evaluating agents under uncommon and nuanced conditions is important, and taxonomy provides a useful organization.

2. Figure 1 suggests strong coverage across both question types and tool types, which is valuable for broad stress testing.

3. The data generation workflow is potentially general and useful. The Planner, Action, and Context Manager roles with iterative DAG updates could transfer to other benchmarks, though the current paper leaves critical details unspecified.

**Weaknesses:**

1. The paper lacks the concreteness needed to verify quality or enable reproducibility. There are no clear descriptions of concrete task families or instances per taxonomy category, and no dataset statistics by difficulty or tool subset. The workflow is described at a high level, but there are no specifics on agent instantiation, prompts, tool lists, or how the “primary” and “weaker” agents are configured. It is not clear how the workflow yields specific benchmark items, how ground truth is defined, or how adversarial feedback is delivered during generation. The reproducibility statement also claims full details are provided, which is not supported by the text.

2. The validity of K-PPL is uncertain. The metric relies on LLM-produced relevance and prior scores, but the implementation details are missing. The same LLM is used to both construct the data and score the DAG, and there is no evidence that an LLM can reliably produce these scores.

3. There are no quality evaluations of the benchmark. The paper does not report human feasibility checks or human baselines, and there is no analysis to ensure that tasks are realistic or solvable by humans. The LLM-as-judge setup lacks agreement studies or rubrics.

**Questions:**

Suggestions to Authors : The paper needs a rewrite. I'm including a checklist of details which is needed for a general benchmark paper:

1. Dataset Examples - Provide examples for each taxonomy category with input, expected output, and validation scripts. Include at least one end-to-end DAG and tool-call trace per example.
2. Dataset Stats - Report dataset statistics. Include counts by category and difficulty, distribution of tools used, and pass rates for baseline agents. Note any time-sensitive tasks and how drift is handled.
3. Task Creation details - Describe agent and LLM instantiations for every workflow step. Release prompts in the appendix, tool inventories and versions, rate limits, and seed diversification procedures.
4. LLM as judge/scorer evaluations - Report human agreement for judging, and provide calibration for K-PPL across models, prompts, and the ω weight, with correlations to human difficulty and solve rates.
5. Benchmark quality checks - Include human annotations for feasibility, correctness, and difficulty, and report a human baseline to anchor performance.
6. Details on evaluation setups - Document prompts, tools, and parameters for all compared agents, and specify how “primary” and “weaker” agents differ.

---

### Meta-Review · Area_Chair_wreN · 2025-12-13

**Summary:**

While the paper addresses an important and timely challenge in evaluating LLM agents under more realistic and demanding conditions, it lacks crucial operational details. The proposed workflow is described primarily at a high conceptual level, making it difficult to assess the quality of the generated tasks or to ensure reproducibility. Moreover, although the K-PPL metric is conceptually interesting, it is not supported by rigorous or quantitative validation.

**Reviewer Concerns:**

No author response was submitted.

**Reviewer Scores:**

No author response was submitted.

---

### Decision · Program_Chairs · 2026-01-26

Reject